# New Insights into the Role of PPARγ in Skin Physiopathology

**DOI:** 10.3390/biom14060728

**Published:** 2024-06-19

**Authors:** Stefania Briganti, Sarah Mosca, Anna Di Nardo, Enrica Flori, Monica Ottaviani

**Affiliations:** Laboratory of Cutaneous Physiopathology and Integrated Center of Metabolomics Research, San Gallicano Dermatological Institute, IRCCS, 00144 Rome, Italy; stefania.briganti@ifo.it (S.B.); sarah.mosca@ifo.it (S.M.); anna.dinardo@ifo.it (A.D.N.); monica.ottaviani@ifo.it (M.O.)

**Keywords:** PPARs, skin physiology, inflammatory skin disease, skin cancer, sebaceous gland, lipids

## Abstract

Peroxisome proliferator-activated receptor gamma (PPARγ) is a transcription factor expressed in many tissues, including skin, where it is essential for maintaining skin barrier permeability, regulating cell proliferation/differentiation, and modulating antioxidant and inflammatory responses upon ligand binding. Therefore, PPARγ activation has important implications for skin homeostasis. Over the past 20 years, with increasing interest in the role of PPARs in skin physiopathology, considerable effort has been devoted to the development of PPARγ ligands as a therapeutic option for skin inflammatory disorders. In addition, PPARγ also regulates sebocyte differentiation and lipid production, making it a potential target for inflammatory sebaceous disorders such as acne. A large number of studies suggest that PPARγ also acts as a skin tumor suppressor in both melanoma and non-melanoma skin cancers, but its role in tumorigenesis remains controversial. In this review, we have summarized the current state of research into the role of PPARγ in skin health and disease and how this may provide a starting point for the development of more potent and selective PPARγ ligands with a low toxicity profile, thereby reducing unwanted side effects.

## 1. Introduction

Peroxisome proliferator-activated receptors (PPARs) are a class of nuclear receptor proteins that act as transcription factors regulating target gene expression. Since their identification in the 1990s [1], PPARs have been the focus of many studies in the literature due to their role in several critical biological processes, such as energy homeostasis and inflammation [2,3,4,5].

PPARs are known to be classically activated by binding to specific ligands, as well as other members of the nuclear receptor superfamily. Following ligand interaction, PPAR undergoes a conformational change that allows it to form heterodimers with the 9-cis retinoid X receptor (RXR) (Figure 1A). The nature of the ligand is important in driving the genomic PPAR response, leading to the recruitment of several co-activators or co-repressors [6].

The resulting complex is able to bind specific DNA sequences in their promoter regions, known as PPAR response elements (PPREs), thereby regulating the expression of target genes involved in a range of physiological processes related to inflammation, metabolism, and cell differentiation [5,7]. In addition to the transcriptional activation and repression, an additional mechanism described in the literature is that PPARs can also act as transrepressors of genes primarily associated with pro-inflammatory pathways through a protein–protein PPRE-independent interaction with other transcription factors, such as NF-kB and AP1 [8,9].

There are three PPAR isoforms, PPARα, PPARβ/δ, and PPARγ, which share structural and sequence homology but differ in tissue distribution, ligand selectivity, and receptor responsiveness, demonstrating their ability to regulate distinct sets of genes [2,10,11,12,13]. The size of the ligand-binding cavities of PPARs is comparable, with about 80% of the amino acids conserved. Nevertheless, amino acid sequence variations between PPARs affect the specificity of the ligands (Figure 2) [14].

All PPAR isoforms are expressed in the human skin and are involved in various cellular functions, including cell proliferation and differentiation [4,18,19,20] (Figure 1B). PPARα and PPARγ are known to induce sebocyte differentiation and are involved in keratinocyte differentiation and epidermal lipid synthesis, with a critical role in epidermal barrier repair [21,22]. In particular, the activation of PPARγ leads to the inhibition of keratinocyte proliferation, promoting terminal epidermal differentiation instead [23,24]. PPARβ/δ is involved in skin wound healing, lipid synthesis, keratinocyte survival, migration, and sebocyte maturation [25,26,27,28]. Several studies in the literature show that activation of PPARα and PPARγ is associated with the inhibition of melanocyte proliferation and stimulation of pigmentation, whereas PPARβ/δ does not yet appear to affect melanocytes [29,30,31,32]. Furthermore, cutaneous immune cells, such as Langerhans and T cells, express all three PPARs isoforms, highlighting their importance in exerting anti-inflammatory effects [4,33,34]. Therefore, the dysregulation of PPAR expression is associated with inflammatory skin disorders and skin malignancies, suggesting their potentiality as therapeutic targets using specific ligands [20].

Typically, PPARs are activated by binding specific ligands that can be natural or synthetic compounds. Fatty acids and their derivatives are examples of endogenous ligands, while synthetic ligands include therapeutic drugs such as thiazolidinediones (TZDs) (PPARγ agonists) and fibrates (PPARα agonists) [2,6,35,36]. In particular, TZDs, such as pioglitazone, and WY14643 are some of the receptor ligands that have demonstrated promise in the management of a range of skin conditions, including psoriasis, atopic dermatitis, and skin cancers [20,28,37,38]. Moreover, it has been discovered that PPARγ agonists decrease skin fibrosis and accelerate wound healing [39,40]. However, PPAR modulators should be carefully evaluated to determine their efficacy and potential side effects [6,37]. Data in the literature show concerns about possible cardiovascular hazards and carcinogenicity associated with PPAR agonist use, especially PPARα/γ dual agonists [41]. TZDs, which are known for their effectiveness in treating type 2 diabetes due to their ability to improve glycaemic control and induce insulin sensitivity, are not good candidates for long-term therapies, such as chemotherapy for skin cancer, due to their lack of specificity and harmful side effects [42,43,44,45,46,47]. Much attention is therefore being paid to the development of new classes of selective and potent PPARγ modulators retaining the beneficial activity of PPARγ agonists while eliminating many of the undesirable side effects [6].

In this review, we have attempted to describe current knowledge of PPARγ in skin physiopathology as the main cutaneous therapeutic target in skin inflammatory disorders and cancer.

## 2. Methods

### 2.1. Three-Dimensional Crystal Structure of PPARs and Sequence Alignment

The 3D structures of hPPARα, hPPARβ/δ, and hPPARγ were obtained by using the US RCSB PDB (RCSB.org) data center for the global Protein Data Bank (PDB) archive of 3D structure data for large biological molecules, with PDBentries of 1I7G [15], 3TKM [16], and 2F4B [17], respectively. Ligands and water in crystal structures were removed (Figure 2A).

The amino acids sequence alignment of the three receptors was obtained by using UniProt Align of hPPARα (Q07869), hPPARβ/δ (Q03181), and hPPARγ (P37231) (UniProt: the Universal Protein Knowledgebase 2002–2024; current release 2024).

### 2.2. Chemical Structures Drawing

The chemical structures of the most representative PPARγ activators (Figure 3) were created using ACD/ChemSketch (Freeware) Version 2022.2.3 Software (Advanced Chemistry Development, Inc., Toronto, Canada).

## 3. PPARγ Modulation and Skin Barrier Homeostasis

The structure and composition of the outermost layer of the epidermis, the stratum corneum (SC), are related to the proper functioning of the skin barrier, mediating the maintenance of water and electrolyte balance and preventing dehydration [48,49]. The SC consists of corneocytes, which are anucleated, terminally differentiated keratinocytes embedded in a lamellar lipid matrix, and provide a barrier to the movement of water and electrolytes [50,51]. The process of differentiation of keratinocytes into corneocytes is also associated with the expression of specific proteins, such as keratin, fillagrin, transglutaminase, involucrin, and loricrin, which have the function of forming the structural scaffolds of the extracellular lipid matrix [50]. Moreover, these proteins are also targets of PPAR activation [52]. In addition, differentiating keratinocytes secrete ceramides, the predominant lipid species in the extracellular matrix, which play an important role in the water-holding and barrier functions of the SC [51]. As epidermal lipid metabolism and skin barrier function are closely linked, modulation of lipid metabolism by PPAR ligands was tested for its effects on the skin barrier. During keratinocyte differentiation induced by high calcium concentration, the levels of PPARγ increase along with the terminal differentiation [18]. In terminally differentiated keratinocytes, PPARγ levels increase 5-fold, peaking in the suprabasal layer [53].

Several studies in epidermal keratinocytes or mice have shown that either the overexpression of PPARγ or its activation by agonists can have beneficial effects on skin barrier function. PPARγ induces a shift in the balance between differentiation and proliferation towards differentiation, leading to the normalization of terminal differentiation of epidermal keratinocytes and a reduction in their proliferation rate [24]. The treatment of animals with PPARγ agonists also reduces the proliferation rate of epidermal keratinocytes, and this anti-proliferative effect is more rapid in recovering epidermis with a disrupted skin barrier [23,54,55]. In mice knockout for epidermal PPARγ, a significant reduction in the transcription of several genes associated with lipid barrier formation has been reported [21,56]. In human epidermal equivalents, barrier disruption mediated by chemical agents such as SDS or acetone resulted in reduced *PPARγ* gene expression levels, suggesting that PPARγ signaling is closely correlated with epidermal barrier fitness [57]. PPARγ ligands may therefore represent useful therapies for various inflammatory skin conditions characterized by epidermal barrier impairment.

## 4. PPARγ and the Sebaceous Gland

The Sebaceous Gland (SG) is an epithelial appendage with an acinar structure consisting of sebocytes; SG is part of the pilosebaceous unit, and its development is closely associated with the formation of the hair follicle (HF) during skin morphogenesis [58,59]. SG expresses all PPAR isoforms. However, PPARγ seems to be a crucial and necessary requirement for SG genesis and homeostasis [60,61,62,63]. Mice with a targeted deletion of PPARγ show the dramatic downregulation of transcripts encoding for genes related to gland development and sebum production, leading to an asebia phenotype. The almost complete ablation of SGs is also associated with a slow hair cycle, altered HF morphology, and skin inflammation, indicating the importance of PPARγ not only for the SG but likely for the whole pilosebaceous unit [21,63,64,65]. Due to its role in SG specification, PPARγ can also be considered a fate determinant for SG cell reprogramming; in fact, the presence of a PPARγ agonist in the culture medium of epidermal stem cells is necessary to induce their differentiation into sebocytes [66]. More recently, it has been proven that the stable overexpression of PPARγ alone is sufficient to drive human keratinocyte conversion into culture-expandible SG cells, with gene expression patterns and functional properties mimicking sebocytes, particularly in terms of lipid synthesis. Furthermore, in mice, the intradermal injection of these “induced SG cells” gives rise to a de novo SG-like structure capable of producing a lipidomic profile closely related to those expected for native SG [63,66].

In addition to its relevance in SG morphogenesis, PPARγ also plays a key role in the maintenance of the SG primary function, namely sebum production, since its expression is associated with sebocyte differentiation. Sebocytes undergo multiple cell state transformations moving toward the sebaceous duct and reach their maturation at the distal end of the SG which constitutes about 20–50% of the SG volume [58,63,67] Variation in the differentiation stage of sebocytes thus characterizes the layered structure of the SG. The peripheral zone is made up of undifferentiated and proliferative sebocytes, whereas differentiating sebocytes, which synthesize and accumulate lipids in cytoplasmic droplets that notably increase their size, constitute the SG maturation zone. Finally, in the necrotic zone, mature and terminally differentiated sebocytes undergo a specialized form of cell death called holocrine secretion, releasing sebum on the skin surface through the HF infundibulum and the sebaceous duct [58,68,69,70]. Together with the increase in lipid production, more differentiated sebocytes of the central zone of the gland highly express PPARγ in contrast to sebocytes present in the SG peripheral zone, underlining the importance of this transcription factor in lipid synthesis and metabolism [62,71,72]. PPARγ activation resulted, in fact, in lipidogenic induction and increased sebum secretion in sebocyte cell lines (SZ95, SEB-1) and human skin, respectively [62,72]. Moreover, PPARγ expression in the different SG zones correlates with the expression of androgen receptors and androgen-metabolizing enzymes, indicating the cooperative role of PPARγ towards androgenic activity. This interplay is also observed in the SZ95 experimental model, where PPARγ induction produced increased sensitivity to testosterone [62,67,73,74,75]. Further investigation of the role of PPARγ in the SG may help to better define the still underestimated and not fully explored SG activity and function.

## 5. PPARγ and Skin Inflammation

Inflammatory responses promote acute or chronic diseases characterized by excessive production of arachidonic acid-derived eicosanoids, inflammatory cytokines, and adhesion molecules [76]. Skin inflammation is a strictly regulated process. PPARs represent one of the main regulators, and their activation by nitrated fatty acid derivatives and cyclopentenone prostaglandins, which are products formed in the late stages of the inflammatory reaction, results in the inhibition of core elements of the inflammatory reaction [9]. Multiple direct and indirect mechanisms promote the anti-inflammatory effects of PPARs, which are involved in the regulation of immune cells and the resolution of skin inflammation [3]. Numerous inflammatory mediators and cytokines are inhibited by PPARγ ligands in various cell types, including monocytes/macrophages, epithelial cells, smooth muscle cells, endothelial cells, dendritic cells, and lymphocytes [77,78,79]. Additionally, PPARγ regulates the biological activities of Langerhans cells [80,81] and decreases the expression of adhesion molecules [82]. PPARγ agonists promote the differentiation of hemopoietic progenitor cells in Langherans cells [83], whereas the pro-inflammatory cytokines inhibit this process. Increased expression of PPARγ affects the maturation and function of Langerhans cells, mainly by accelerating lipid metabolism [84] and fatty acid oxidation [85]. In addition, the PPAR-signaling pathway enhances immunogenicity and T cell priming by Langerhans cells [86]. PPARγ is involved in the differentiation and proliferation of T cells and plays a crucial role in their activation following antigen recognition, causing rapid changes in their phenotype. A shift from a resting state to a state with a much higher metabolic demand follows stimulation of the T cell receptor, activating, in turn, PPARγ and inducing the genes involved in glucose and fatty acid uptake [87,88]. The activated CD4+ T cells differentiate into several subpopulations with different inflammatory and metabolic phenotypes, namely Th1, Th2, Th17, and Treg [89]. Various PPARγ agonists reduce proliferation, inhibit Th1 and Th17 differentiation, and promote Treg differentiation of CD4+ T cells [33,90,91].

In macrophages, PPARγ regulates polarization, maturation, epigenetics, and metabolism. In particular, PPARγ modulates the polarization of pro-inflammatory M1 macrophages into anti-inflammatory M2 macrophages [92]. In addition, after stimulation with LPS, a condition that drives M1 polarization, macrophages knockout for PPARγ produce high levels of pro-inflammatory cytokines, such as IL-1, which are considered markers of M1 polarization [79]. PPARγ also plays a role in CD36 expression, oxLDL uptake, and foam cell formation by macrophages [93]. PPARγ deficient macrophages have, in fact, diminished capacity to uptake and degrade OxLDL [94]. Notably, not only cholesterol uptake but also its efflux is under the transcriptional control of PPARγ through the cassettes of the ABCA1 family [94]. Furthermore, the overexpression of dominant-negative PPARgamma in macrophages results in the upregulation of pro-inflammatory cytokines/chemokines and expansion of myeloid-derived suppressor cells (MDSCs), leading to inflammation, immunosuppression, and tumorigenesis [95]. These data suggest that the PPARgamma signaling pathway and its downstream gene products are essential for controlling chronic inflammation (especially MDSC homeostasis) and tumorigenesis.

PPARγ is a key regulator of the functional maturation of dendritic cells (DC), driving the induction of immunogenic T cell responses versus immune tolerance. Klotz et al. demonstrated that pharmacological modulation of PPARγ signaling in murine DC reduced the expression of costimulatory molecules and IL-12 associated with the maturation process and significantly inhibited the ability of DC to prime naive CD4+ T cells in vitro [96]. In addition, CD4+ T cells primed by PPARγ-activated DCs failed to express Th1 and Th2 cytokines and did not respond to further T cell receptor-mediated stimulation with secondary clonal expansion [96]. The authors proposed that PPARγ controls DC function in a complex manner that allows the survival of Ag-reactive CD4+ T cells but induces CD4+ T cell anergy instead of immunity upon secondary Ag encounter. Conversely, PPARγ ablation increased DC immunogenicity, suggesting that PPARγ may act as a constitutive regulator of DC suppression [96]. The activation of PPARγ with 15d-PGJ2 or troglitazone inhibits Toll Like Receptor-mediated activation of the MAP kinase and NF-kB pathways and results in a reduced capacity of DC to stimulate T cell proliferation. This highlights the inhibitory effect of PPARγ activation on DC maturation [97]. Through its ability to reduce the activity of transcription factors, such as AP-1, STAT, NF-kB, and NFAT, PPARγ negatively regulates inflammatory gene expression in skin immune cells [98,99]. The formation of inhibitory complexes between PPARγ and the aforementioned transcription factors is called trans-repression [100] and leads to the suppression of pro-inflammatory cytokine genes and the downregulation of cyclooxygenase-2/COX-2/PTGS2 expression, thereby reducing the production of prostaglandins [101]. Due to its anti-inflammatory effects, PPARγ has been the subject of numerous studies in recent years to elucidate pathogenetic mechanisms and develop new therapies for inflammatory skin diseases.

## 6. PPARγ Agonists/Modulators in Skin Disease Management

### 6.1. Atopic Dermatitis and Contact Dermatitis

Atopic dermatitis (AD) is a chronic relapsing inflammatory skin disease, which may be associated with IgE-dependent diseases like allergic asthma, food allergy, and allergic rhinitis. AD patients have skin barrier impairment within crusted erythematous areas, associated with epidermal hyperplasia, scaling, and lichenification [102]. Loss-of-function mutations in the filaggrin gene are a major predisposing factor for the manifestation of AD, increasing the permeability of the skin [103]. However, not only do patients with mutations show decreased filaggrin levels in the skin but inflammatory processes in general also decrease filaggrin expression in people not carrying the respective mutation [104]. Skin injury often triggers AD, allowing increased penetration of environmental allergens and inducing a cytokine milieu that favors a Th2 response [52]. Although information on PPARγ in AD is conflicting, the systemic administration of PPARγ ligands improved experimental skin allergy in mice [105] and the intensity of clinical symptoms in patients with severe AD [106]. Despite the evidence that AD patients may benefit from PPARγ agonists, the clinical observations derive from a small number of patients, so the potential role of PPARγ modulation in the treatment of AD should be evaluated in future controlled trials.

PPARγ agonists may maintain mast cell homeostasis by inhibiting the maturation of their precursors, reducing mast cell phenotypic markers and viability, inhibiting degranulation, and inducing cell apoptosis [107]. These data suggest that PPARγ ligands may serve as effective anti-inflammatory reagents in the treatment of mast cell-related diseases, such as allergic reactions and contact dermatitis. However, recent data report that PPARγ is essential for the promotion of ‘type 2’ immune responses that are typically associated with allergic diseases, highlighting the contrasting role of PPARγ in allergic inflammation [108]. The high availability of PPARγ ligands in the environment is thought to play a key role in the increase in allergic diseases worldwide [108]. This effect is mainly observed with PPARγ ligands from environmental pollutants. The discovery and testing of compounds that can bind PPARγ but primarily activate anti-inflammatory responses may be a valid strategy to counteract or mitigate the negative effects of environmental PPARγ activators. Remarkably, 50% of children with AD develop asthma later in life, and the risk of developing hay fever is up to 75% [109]. This typical sequence of allergic manifestations is called the “atopic march” [110,111]. Glucocorticoids (GCs) are the standard anti-inflammatory drugs used to treat AD and attempt to halt the atopic march. However, GCs have detrimental side effects on the integrity of the epidermal barrier [112,113,114,115]. Therefore, the combination of GCs with therapies, such as PPARγ ligands, that have an anti-inflammatory effect while maintaining the integrity of the skin barrier is an interesting topic for study [115]. The activation of the glucocorticoid receptor (GR) has been shown to contribute to the anti-inflammatory effects of PPARγ activity [116]. Deckers et al. developed a murine model that mimics the atopic march. In this model, skin inflammation is associated with a mixed Th2/Th17 phenotype and severe airway inflammation induced by dust mite challenges. Combined activation of GR/PPARγ suppresses local skin inflammation to a greater extent than single activation but appears insufficient to prevent the allergic immune response in the lung, even though the severity of asthma is effectively reduced by counteracting the Th17 response [117]. Thus, GR/PPARγ co-activation represents a potent remedy against allergic skin inflammation and worsening of atopic march.

The beneficial effects of PPARγ modulators in AD therapy are, therefore, mostly related to in vitro/in vivo studies with cell cultures and animal models, whereas conflicting data are reported in patients. Therefore, controlled clinical studies are needed to test the therapeutic efficacy of PPARγ modulators.

### 6.2. Psoriasis

Psoriasis is a multifactorial inflammatory skin disorder, generally considered a polygenic, immune activation-dominated disease characterized by keratinocyte hyperplasia and disturbed differentiation [118]. The interaction between psoriatic keratinocytes and cells of the immune system is a crucial element in the development of psoriatic lesions since keratinocytes produce chemokines to attract T cells to the inflammatory site [119]. Due to their pro-differentiating, anti-proliferative, and immunomodulatory effects, PPARγ may play a central role in the pathogenesis and treatment of psoriasis. In psoriatic lesional skin, the expression of PPARγ is decreased [98,120]. In a 3D tissue-engineered psoriatic skin model characterized by the reduced expression of PPARγ, the addition of DHA, a long-chain n-3 polyunsaturated fatty acid, regulates the differentiation of psoriatic keratinocytes and promotes the synthesis of anti-inflammatory lipid mediators through its ability to activate the PPARγ pathway [121]. According to the studies performed on mice, either the overexpression of PPARγ or its activation by agonists may potentially produce variable beneficial effects on the skin. In skin hyperproliferative disease murine models, the treatment of animals with PPARγ agonists (troglitazone, rosiglitazone, pioglitazone, and BP-1107) suppresses inflammatory signals, reduces the proliferation rate, and normalizes terminal differentiation of epidermal keratinocytes [23,122,123]. A disruption of fatty acid metabolism and associated PPARγ signaling was revealed by whole transcriptome sequencing [124] and mRNA expression profiling [125] of psoriatic lesions and adjacent normal skin. The PPARγ signaling pathway is proposed as a key regulatory factor in the pathogenesis and a potential therapeutic target of psoriasis. Sobolev et al. combined experimental results and network functional analysis to reconstruct the model of downregulated PPARγ signaling in psoriasis and tested the hypothesis that low PPARγ levels may alter the activity of cellular signaling pathways in the skin, facilitating the chronic inflammatory and immune response in human psoriatic lesions [120]. They found higher expression of *STAT3*, *RORC*, *FOXP3*, and *IL17A* genes overall with low *PPARγ* mRNA levels in psoriatic lesions. Their data support the hypothesis that PPARγ attenuates the expression of genes involved in the development of psoriatic lesions and show that the regulation of PPARγ expression by FOSL1 and by the STAT3/FOSL1 feedback loop may be central in psoriatic skin and T cells [126]. Based on this evidence, the modulation of PPARγ may have therapeutic potential in psoriasis. Several clinical studies show that PPARγ agonists are effective in normalizing the key morphological features of psoriasis. Patients receiving systemic treatment with TDZs troglitazone or pioglitazone showed reduced hyperplasia and normalized histological features of lesional skin [23,127,128,129,130,131]. However, oral rosiglitazone was not more effective than placebo in patients with moderate to severe chronic plaque psoriasis [132]. Topical rosiglitazone was also ineffective [133], suggesting that differences in the ability to modulate PPARγ may explain the different therapeutic responses. In patients with psoriasis, PPARγ agonists may also have an additive effect in combination with typical antipsoriatic drugs. Pioglitazone improved the effect of acitritin [134], methotrexate [135,136], or other antipsoriatic drugs [137] and increased the effectiveness of phototherapy in eligible psoriasis patients [138]. However, like many other drugs, TZD therapies have unwanted side effects due to their simultaneous action on different cells and non-specific off-target effects. TZDs promote adipogenesis and adipocyte maturation, leading to adipocyte hypertrophy, and induce edema, weight gain, and increased risk of cardiovascular events and bone fractures in patients [98]. The new generation of PPARγ agonists is expected to retain the beneficial properties of TZDs while significantly reducing the disruption of lipid and glucose metabolism, which is actively involved in the side effects observed in patients. A recently discovered non-TZD agonist of PPARγ, GED-0507-34L, binds to PPARγ with relatively high affinity and reduces the inflammatory response by inhibiting NF-kB and upregulating the expression of IkBα, as well as suppressing the prostaglandin-endoperoxide synthase 2 (PTGS2) in normal human epidermal keratinocytes and lymphocytes [101]. In IL21-induced psoriasis-like skin lesions in mice, topical application of GED-0507-34L reduces the accumulation of cellular infiltrate and epidermal hyperplasia, suggesting that this PPARγ modulator may potentially benefit patients with psoriasis [101]. A Phase I clinical trial in psoriasis patients (Reg. No.: GED0507-PSO-01-12, USA) showed that, unlike TZDs, GED-0507-34L had no serious side effects other than redness and itching in the treated areas (in 1 of 24 treated patients) [98]. However, despite the increasing development of new agonists, the potential benefits of new PPARγ modulators in the treatment of psoriasis need to be confirmed in clinical trials to ensure that the increase in efficacy is balanced by a reduction in potential side effects in long-term therapy.

### 6.3. Vitiligo

Vitiligo is a relatively common skin disorder characterized by a reduction in the number and function of melanocytes, resulting in hypopigmented or depigmented skin lesions affecting 0.5–1% of the world’s population. Several factors including genetics, autoimmunity, oxidative stress, melanocyte apoptosis, and neurological mechanisms have been implicated in the pathogenesis of vitiligo [139]. The destruction of melanocytes within vitiligo lesions is mediated by the differentiation of DCs, leading to the transformation of CD4+ T cells into Th1 or Th17 lymphocytes, which secrete various pro-inflammatory cytokines such as IFN-γ, TNF-α, and IL-17, further damaging melanocytes and activating B lymphocytes to produce antibodies against autoantigens such as tyrosinase, tyrosin hydroxylase, and Sox10 [73,140,141,142]. Simultaneously, CD8+ T cells are activated and exert their destructive effect on melanocytes [143]. Improving the understanding of the inflammatory pathways in the pathogenesis of vitiligo and identifying more therapeutic methods to promote the proliferation and function of melanocytes has been a hot topic in the field of vitiligo treatment. In a recent study, a comprehensive bioinformatics analysis was conducted to explore potential genes and signaling pathways associated with vitiligo and metabolic diseases [144]. PPARγ pathway genes were significantly downregulated in vitiligo samples, and the drug–gene interaction analysis suggested that the PPARγ agonist rosiglitazone may activate the *EDNRB* gene expression to enhance melanogenesis. Thus, exploring the role of the PPARγ pathway may provide new insights into the pathogenesis and discovery of potential therapeutic targets of vitiligo. Treatment of vitiligo melanocytes and fibroblasts with pioglitazone ameliorates mitochondrial alterations by reducing ROS generation, restoring mitochondrial membrane potential, increasing ATP levels, and increasing anaerobic glycolytic enzyme expression and protein levels [145]. In the same study, pioglitazone reverses premature melanocyte senescence in vitiligo. Recently, pioglitazone was found to ameliorate defective differentiation/barrier constitution and reduce inflammatory mediator production in vitiligo keratinocytes [146]. This suggests that PPARγ activation may preserve and enhance skin barrier homeostasis in vitiligo patients, potentially counteracting the effects of environmental stressors and subsequent triggering of inflammation. Further clinical studies are needed to investigate the efficacy of selective modulators of PPARγ as a therapy to treat and/or slow the clinical manifestations of vitiligo and its spread.

### 6.4. Acne

Acne is one of the most common SG-related inflammatory disorders, in which the altered SG activity, characterized by both quantitative and qualitative sebum alterations, plays a pivotal role. This abnormal sebum production contributes to and affects the other pathogenetic factors, namely follicular hyper-keratinization, inflammation, and *C. acnes* colonization, representing the driving multifactorial process of the disease. Due to its role in sebocyte differentiation and lipid production, PPARγ has been considered a possible target for the treatment of acne [70,147,148]. In addition to PPARγ, SG activity is also regulated by insulin and insulin-like growth factor (IGF1), indicated by growing evidence as important hormonal triggers in acne development [149,150,151,152,153,154,155]. In SZ95 sebocytes, insulin stimulates lipogenesis, as well as inflammation, through PI3K-Akt-mTOR signaling. As far as lipid induction is concerned, insulin challenge gives rise to qualitative lipid composition mimicking acne sebum in terms of eicosanoid production, and thus in terms of lipo-inflammatory process driven by eicosanoid-producing enzymes, such as lipoxygenase (LOX) [22,67,148,156,157]. The magnitude of the insulin effect depends on the sebocyte differentiation stage. In fact, the reduced expression of PPARγ in low differentiated SZ95 sebocytes is associated with a higher level of both insulin and insulin-like growth factor-1 receptors (IR, IGF1R), and with the consequent up-modulation of Akt-mTOR signaling, leading to higher responsiveness of these cells to insulin stimulus in comparison with more differentiated sebocytes. Furthermore, the reduced level of PPARγ may also be associated with a higher inflammatory reaction after insulin stimulation due to the increase in LOX activity, which can also result in the production of IL6 and IL8 cytokines, beyond eicosanoid increase, in sebocytes [22,148,156]. Accordingly, in vivo data in acne skin demonstrated lower levels of PPARγ together with increased AkT-mTOR signaling as well as higher 5-LOX expression and eicosanoid levels [158,159,160]. The PPARγ modulation by the selective and newly designed agonist N-Acetyl-Ged-0507-34-Levo (NAC-GED0507; GMG-43AC) is able to counteract the acne-like altered lipogenesis and inflammatory process produced by insulin treatment in SZ95 sebocytes. The induction of sebocyte differentiation and thus the increase in PPARγ expression by NAC-GED is considered the basis for the correct sebocyte response to insulin stimulation, leading to improved sebum composition. In addition, parallel anti-inflammatory effects of PPARγ were observed through the reduction in LOX activity and antagonism of the NF-kB pathway [22,156]. The clinical efficacy of topical NAC-GED0507 gel was demonstrated and then confirmed by a phase 1 and phase 2 trial, respectively. The knowledge gained so far identifies PPARγ modulation as a key element to be taken into account in novel and effective therapeutic strategies for acne treatment [156,161].

### 6.5. PPARγ and Skin Aging

Aged skin is characterized by structural and functional changes in the dermis, which include the formation of lines and wrinkles, increased pigmentation, loss of elasticity and firmness, and gray skin [162,163]. Skin aging is the hallmark of prolonged UV exposure, which causes photoaging, and intrinsic aging-related factors, such as collagen breakdown, by increasing the expression levels of metalloproteinase (MMP) enzymes [164,165,166]. Endogenous systems for photoprotection include epidermal thickening, pigmentation, and complex antioxidant and DNA repair systems [167,168,169]. Excessive production of ROS is known to be an initiator and promoter of intrinsic aging and photoaging [170]. UV-induced skin inflammation is associated with both intrinsic and extrinsic skin aging processes through the release of pro-inflammatory cytokines by the accumulation of ROS [171,172,173]. Since intracellular oxidative stress is mainly related to the decline of endogenous antioxidants, such as reduced glutathione and catalase with UV exposure and aging, treatment with exogenous antioxidants would be an effective approach to prevent the progression of skin aging [174,175]. The PPAR family, through its ability to modulate inflammatory response and antioxidant defense, can modulate the balance between MMP activity and collagen expression to maintain skin homeostasis and counteract skin aging. Several studies have suggested that PPARγ represents a therapeutic target for the photoaging process, as it is able to inhibit the expression of MMPs and AP-1 and modulate oxidative stress-sensitive pathways [176,177]. Intensive studies of PPARα/γ dual modulators have revealed their importance in age-related inflammation and photoaging as regulators of cytokines, MMPs, and NF-kB [178,179,180]. Decreased PPARγ activity is closely related to chronic inflammation associated with the aging process in vivo. Pioglitazone treatment contributes to attenuating several age-related disorders in aged apoE-/- mice, thereby representing a promising protective therapy against aging and age-related diseases [181]. Aging impairs epidermal re-growth during wound healing and results in lower expression of peroxisome proliferator-activated receptor gamma coactivator-1 alpha (Pgc-1a) [182]. In addition, it has been reported that PPARγ modulates UVB-induced COX-2 expression and subsequent synthesis of PGE2, a lipid mediator that suppresses collagen synthesis in dermal fibroblasts by partially interfering with TGF-β signaling [183,184]. In UVB-exposed human keratinocytes, chimyl alcohol reduces cellular damage by suppressing *COX-2* mRNA expression and PGE2 synthesis and stimulates the intracellular defense system against ROS through Nrf2 signaling activated by a PPAR-γ-dependent mechanism [185]. 3-O-laurylglyceryl ascorbate, an amphipathic derivative of ascorbic acid, scavenges intracellular ROS and stimulates intracellular antioxidants such as GSH through the PPARγ and Nrf2 signaling pathways [186].

The ability of PPARγ modulators to counteract the photo-aging process has been extensively studied in experimental models of cell senescence. Due to the key role of oxidative stress in the photoaging process, the transformation of proliferating skin cells into photoaged cells under conditions of artificially elevated ROS levels resembles premature senescence. Therefore, stress-induced premature senescence (SIPS) models can be useful tools to investigate the biological and biochemical mechanisms involved in photo-induced skin damage. PPARγ activation has been demonstrated to reduce age-related inflammation and aging progression in an H_2_O_2_-induced SIPS cell phenotype [187]. In human dermal fibroblasts induced to SIPS by PUVA (a single sub-cytotoxic exposure to UVA in the presence of activated 8-methoxypsoralen), a decrease in PPARγ expression and activity is an early event associated with the induction of a senescence-like phenotype [188,189]. Azelaic acid, a nine-carbon linear saturated dicarboxylic acid, and 2,4,6-octatrienoic acid, a member of the parrodiene family, increase the transactivation of PPARγ and counteract PUVA-SIPS markers, including long-term growth arrest, flattened morphology, increased synthesis of MMPs, increased degradation of ECM, and senescence-associated beta-galactosidase staining, and also interfere with ROS-dependent cellular signaling mechanisms [188,189]. In a reproducible photo-damaged model of mouse dermal fibroblasts treated with repeated UVB exposures, rosiglitazone counteracts the photoaging process [190]. Taken together, PPARγ modulation may counteract both intrinsic and extrinsic skin aging processes. However, further investigation is needed to develop new therapeutic or preventive approaches with oral or topical administration of PPARγ modulators while limiting potential side effects such as lipid accumulation due to increased adipocyte differentiation.

## 7. PPARγ and Skin Cancer

Over the past twenty years, considerable progress has been made in understanding the role of PPARγ in skin cancer. UV exposure is the major cause of skin carcinogenesis. This process involves DNA damage and alterations in signaling pathways essential for cell proliferation, survival, oxidative stress, inflammation, immunosuppression, differentiation, remodeling, and apoptosis [191,192,193].

### 7.1. Non-Melanoma Skin Cancer (NMSC)

The ability of PPARγ activation to induce terminal differentiation, afford photoprotective effects, and inhibit cell growth and inflammation has been demonstrated in several in vitro studies using malignant skin cell types and in animal studies using murine skin tumor models, suggesting that PPARγ agonists may act as tumor suppressors.

The study by Sahu et al. provides evidence that epidermal PPARγ plays a protective role in suppressing UVB-induced NMSC formation and progression in mice. Indeed, mice lacking epidermal PPAR (PPARγ -/-^epi^ mice) show a marked increase in photocarcinogenesis associated with UVB-induced apoptosis, inflammation, barrier dysfunction, and epidermal hyperplasia [56]. A marked increase in TNF-α expression has been demonstrated in the same mouse model. In addition, the PPARγ agonist rosiglitazone reverses the systemic immunosuppression induced by chronic UVB [21]. Park et al. show the protective effects of a PPARα/γ dual agonist on inflammatory responses, epidermal thickness, and lipid peroxidation associated with chronic UVB exposure [194]. Furthermore, mice lacking the PPARγ heterodimerization partner RXRα in epidermal keratinocytes show increased apoptosis, altered epidermal proliferation, and augmented DNA damage in response to UVB exposure [195]. The studies by Ren and Konger and Balupillai et al. support the idea that PPARγ activation may be a potential pharmacological target for the prevention or treatment of skin cancer by suppressing tumor-promoting chronic inflammation associated with UV exposure, as well as by enhancing antitumor immune responses in the skin [196,197]. More recently, reports on human epidermoid carcinoma cells demonstrate the ability of PPARγ ligands to inhibit cell growth by regulating the expression of cell cycle-associated proteins and inducing terminal differentiation [198,199,200]. These results are consistent with the observed reduced anchorage-dependent clonogenicity and marked tumor volume inhibition in ectopic xenografts [199].

The activation of PPARγ by the parrodiene derivative 2,4,6-octatrienoic acid (Octa) prevents cutaneous UV damage by acting on the different skin cell populations [189,201]. Most interestingly, Octa promotes antioxidant defense, greater cell survival, and the effective removal of UV-induced DNA damage in human keratinocytes and human epidermal skin equivalents [202]. In line with our results, Babino et al. proved the good tolerability and long-term efficacy of a medical device containing Octa and urea in the reduction in grade III actinic keratoses (AKs) [203], skin lesions resulting from chronic and excessive UV exposure with a certain risk of becoming cancerous [204,205]. In addition, topical application of sunscreen containing inorganic filters and Octa has been shown to protect against the formation of sunburn cells, reduce the number of apoptotic keratinocytes, and prevent UV-induced molecular alterations [206]. More recently, Flori et al. demonstrated the ability of Octa and its derivative A02 to antagonize, through PPARγ activation, the TGF-β1-mediated changes associated with the epithelial–mesenchymal transition (EMT) process in a human squamous cell carcinoma cell line. Most interestingly, both compounds counteracted the TGF-β1-mediated activation of EMT-related signaling pathways, cell migration capacity, cell membrane lipid remodeling, and the release of bioactive lipids involved in EMT. In addition, in vivo experiments in a mouse model with cutaneous papillomas showed that the same PPARγ ligands, applied topically, were able to reduce the number of lesions and tumor area [200].

In terms of human epidemiological evidence, a large population-based retrospective cohort study in Taiwan suggests that the long-term use of the PPARγ agonist rosiglitazone in patients with type 2 diabetes mellitus may be associated with a lower risk of NMSC [207]. Unfortunately, there is little direct evidence from clinical trials on the feasibility of using PPARγ modulators for the prevention and/or treatment of keratinocyte-derived cancers. However, data from the literature strongly support the hypothesis that targeting PPARγ may represent a potential pharmacological target for the prevention or treatment of skin cancer. Current knowledge could be considered as a starting point for developing more potent and selective chemopreventive or chemotherapeutic PPARγ ligands with a low toxicity profile, thus reducing unwanted side effects.

### 7.2. Melanoma

Some evidence in the literature suggests that PPARγ may be involved in protecting against melanoma, although current knowledge is limited and controversial. Similar to non-melanoma skin cancers, growth inhibition via PPARγ activation has been described in melanoma. Most of these studies used melanoma cell lines and treatment with PPARγ agonists in vitro [193]. The study by Botton et al. shows that the PPARγ ligand ciglitazone inhibits melanoma cell growth by inducing both cell cycle arrest via a PPARγ-dependent pathway at low concentrations of the compound and apoptosis at higher concentrations and independently of PPARγ activation. Furthermore, in vivo treatment of nude mice with ciglitazone significantly impairs the development of human melanoma xenografts [208]. In addition, the PPARγ ligand 15-deoxy-D12,14 prostaglandin J2 (15d-PGJ2) has been shown to induce cell cycle arrest in several melanoma cell lines and to inhibit tumor cell migration, tumor-associated fibroblast proliferation, and endothelial cell tube formation [209]. A connection between the α-melanocyte stimulating hormone (α-MSH) and PPARγ in human melanocytes and melanoma cells has been identified, resulting in reduced melanoma cell proliferation associated with cell cycle withdrawal [32,210,211]. In line with these findings, Borland et al. demonstrated that overexpression and/or activation of PPARγ by rosiglitazone inhibits cell proliferation and anchorage-dependent clonogenicity in a human melanoma cell line. This effect is reflected in the observed inhibition of ectopic xenograft tumorigenicity [199]. Moreover, Konger et al. demonstrate that B16F10 melanoma tumor growth is enhanced in syngeneic PPARγ -/-^epi^ mice, indicating that loss of epidermal PPARγ acts through indirect mechanisms to regulate tumor growth. Moreover, the pretreatment of mice with rosiglitazone counteracts B16F10 tumor growth induced by UVB [212]. A clinical study on patients with late metastatic stage melanoma shows a significantly prolonged progression-free survival in the group of patients that received chemotherapy in combination with pioglitazone compared to the group that received chemotherapy alone [213]. In contrast to these literature data showing a potential efficacy of PPARγ activation in the prevention or treatment of melanoma, a cohort study conducted in diabetic patients treated with pioglitazone suggests that the use of the TZD class of PPARγ ligands is associated with an increased risk of melanoma [214]. In line with this evidence, Pich et al. show that rosiglitazone induces a tumorigenic paracrine communication program in a subset of human metastatic melanoma cells involving the secretion of cytokines, chemokines, and angiogenic factors, leading to the activation of non-malignant stromal cells of the tumor microenvironment. Moreover, in vivo data show that a rosiglitazone-supplemented diet promotes melanoma tumor growth in xenografts, accompanied by increased inflammation and angiogenesis [215]. The role of PPARγ agonists in tumourigenicity is complex [216]. The co-occurrence of data suggesting both beneficial and tumorigenic effects of PPARγ ligands indicates a great complexity in the mechanisms of PPARγ action in relation to the tumor context and microenvironment. Therefore, the role of PPARγ in melanoma remains to be elucidated. Further clinical studies are needed to determine whether PPARγ modulators may have a beneficial role in the prevention and/or treatment of melanoma. Therapeutic modulation of PPARγ should, therefore, be considered with caution.

## 8. Conclusions

Over the past 20 years, considerable progress has been made in understanding the role of PPARγ in skin physiopathology. Keratinocyte differentiation, barrier permeability maintenance, sebaceous gland development and function, and inflammatory response are all processes regulated by PPARγ transcriptional activity. Alteration and/or disruption of these physiological functions are associated with the onset of skin disorders. The development of drugs targeting PPARγ is an area of active research, with PPARγ agonists being considered potential future dermatology therapeutics (Table 1; Figure 3).
Figure 3Representative PPARγ activators. Chemical structures of the most tested PPARγ activators related to skin pathophysiology. Thiazolidinedione (TZD).
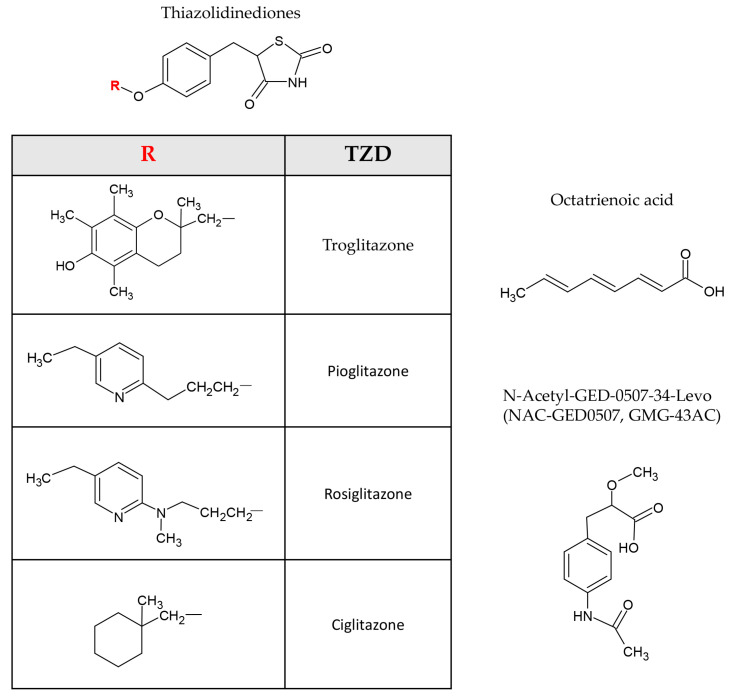



A major challenge for future research will be the development of novel PPARγ ligands capable of selectively activating the different patterns of genes involved in the desired beneficial activity while reducing unwanted side effects through the design of ligands with a low toxicity profile. Much work is needed to fully characterize the pharmacological activity and specificity of these compounds.

## Figures and Tables

**Figure 1 biomolecules-14-00728-f001:**
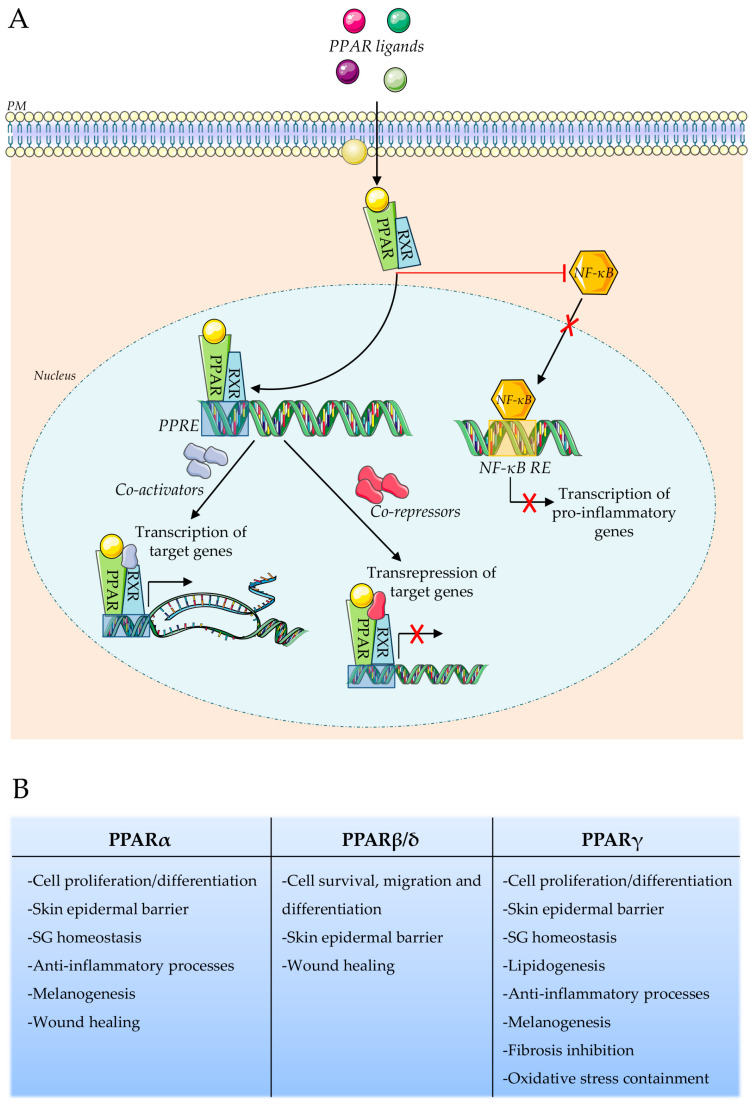
PPARs signaling and functions. (**A**) Schematic representation of the ligand-dependent PPARs activation. The ligand nature leads to the recruitment of several co-activators or co-repressors, important in driving the PPAR genomic actions. Cytoplasmic PPARs, by interacting with other transcription factors, such as NF-kB, elicit non-genomic effects by negatively regulating pro-inflammatory gene expression (**B**) Principal roles exerted by the different PPARs isoforms in skin homeostasis.

**Figure 2 biomolecules-14-00728-f002:**
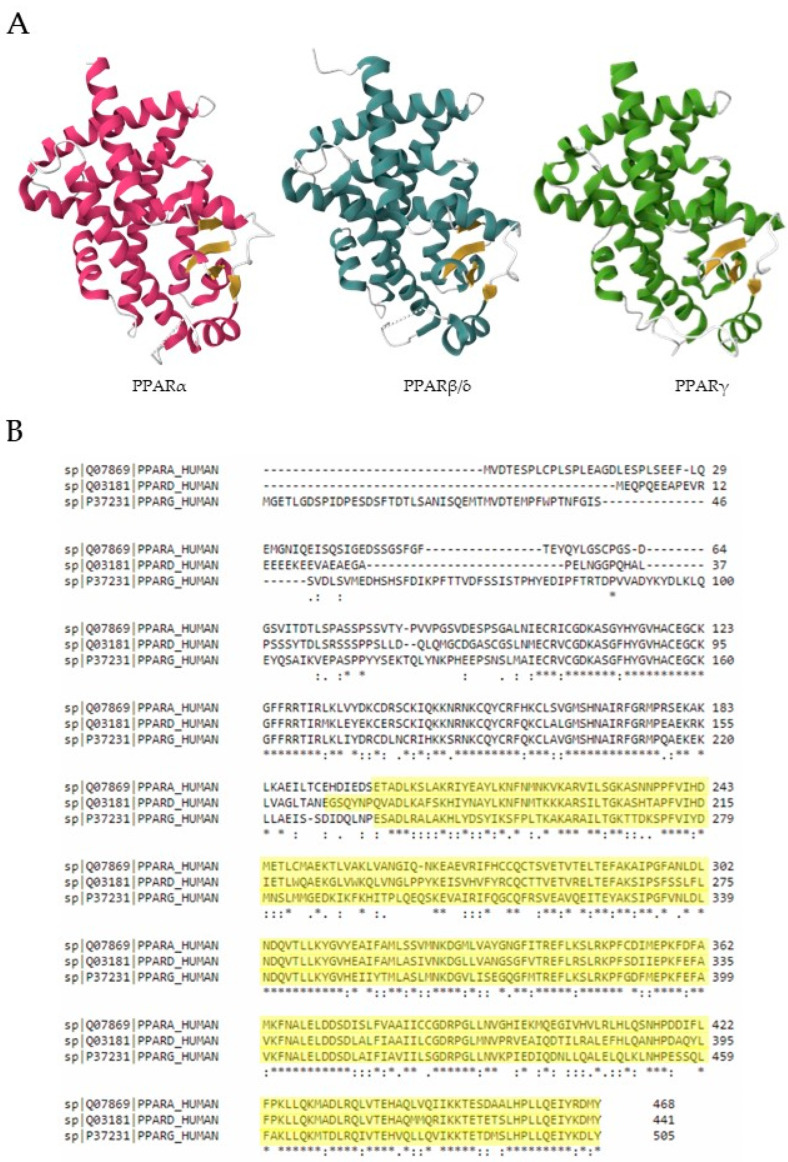
Three-dimensional structure and ligand binding pocket aminoacid sequences of three PPAR isoforms. (**A**) 3D structure of PPARα (1I7G [15]), PPARβ/δ (3TKM [16]), and PPARγ (2F4B [17]). (**B**) Structure-based sequence alignment of hPPARα (UniProt ID Q07869), hPPARβ/δ (UniProt ID Q03181), and hPPARγ (UniProt ID P37231). For each aligned residue pair, “:” is for conserved substitutions, “.” denotes semi-conserved substitutions, and “*” denotes identical residues. Residues binding ligands are highlighted in yellow.

**Table 1 biomolecules-14-00728-t001:** Therapeutic targeting of PPARγ in skin diseases. Beneficial effects of PPARγ activation in various skin disorders, listing the most tested molecules and the corresponding references.

Skin Condition	PPARγ Beneficial Effects	PPARγ Activators	References
Atopic Dermatitis, Contact Dermatitis	- Promotion of anti-inflammatory responses - Maintenance of mast cell homeostasis - Maintenance of barrier integrity and homeostasis	pioglitazone, ciglitazone	[107,115,116]
Psoriasis	- Reduction in the expression of genes involved in the development of psoriatic lesions - Normalisation of lesional skin - Reduction in epidermal hyperplasia	troglitazone, rosiglitazone, pioglitazone, BP-1107, GED-050734L	[23,98,101,122,123,124,125,126,134,135,136,137,138]
Vitiligo	- Enhancement of melanogenesis - Reduction in mitochondrial alterations - Reduction in ROS generation - Reversal of premature melanocyte senescence - Improvement of defective differentiation/barrier constitution	rosiglitazone, pioglitazone	[145,146]
Acne	- Reduction in altered lipogenesis - Anti-inflammatory activity - Improvement of sebum composition	NAC-GED0507	[22,156,161]
Skin Aging	- Modulation of aged-related inflammation by inhibition of pro-inflammatory gene expression - Reduction in aging progression and age-related disorders	pioglitazone, azelaic acid, kojyl cinnamate ester derivatives, octa, rosiglitazone	[177,181,188,189,190]
Non-melanoma skin cancer	- Antioxidant defense - Reduction in UV-dependent DNA damage - Inhibition of cell growth and induction of terminal differentiation - Antagonization of the EMT process	rosiglitazone, octa, A02	[21,198,199,200,202]
Melanoma	- Inhibition of cell growth - Induction of cell cycle arrest - Inhibition of tumor cell migration - Reduction in melanoma cell proliferation	ciglitazone,15d-PGJ2, rosiglitazone, pioglitazone	[193,199,209,212,213]

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
