# Peer review of "New Insights into the Role of PPARγ in Skin Physiopathology"

_biomolecules, 2024, doi:10.3390/biom14060728_

Round 1

Reviewer 1 Report

Comments and Suggestions for Authors

This review by Briganti et al summarizes multiple studies that indicate an important role for PPARgamma in skin disease and physiology.  After briefly describing the role of PPARgamma as a ligand-activated nuclear receptor, the review discusses the role of PPARgamma in normal skin physiology, including barrier function, sebaceous gland development and corneocyte lipid synthesis.  The review then discusses the role of PPARgamma in skin inflammation, then follows this with a discussion of PPARgamma in skin diseases, particularly inflammatory skin disorders, aging and cancer.  Overall, the review provides a very nice summary of the evidence linking PPARgamma activity in skin biology and common skin diseases.  There are several issues that should be addressed:

1.  Much of the review focuses on the ability of PPARgamma to act as an anti-inflammatory agent.  This anti-inflammatory activity is likely dependent on the ability of PPARgamma to transrepress pro-inflammatory nuclear transcription factors.  While this transrepressive activity is briefly mentioned, it would be useful to include a greater discussion of the mechanisms of transrepression.  A second figure, similar to Fig 1 that highlights transrepressive mechanisms might also be useful. 

2. Page 5 discusses the effects of PPARgamma on macrophages.  It would be useful to include studies by Hong Du and Cong Yan that show that dominant negative PPARgamma expression in macrophages drives MDSC formation, immunosuppression and tumorigenesis [Blood. 2012, 119(1):115-26].  This study is relevant to both macrophage biology as well as the later discussion of PPARgamma in cutaneous malignancy.   

3. Page 5, lines 200-2001. "PPARg is a key regulator of the functional maturation of dendritic cells (DC), driving the induction of immunogenic T cell responses rather than immune tolerance".  Yet the next sentence would suggest the opposite: "Pharmacological modulation of PPARg signaling in murine DC ...... significantly inhibited the ability of DC to prime naive CD4+ Tc cells in vitro [91]."  The study referenced for this statement also indicates that adoptive transfer of PPARg-activated DCs induced CD4+ T cell anergy.  How does this indicate that PPARgamma drives the ability of DCs to induce immunogenic T cell responses?  

Minor issues:

1. Page 3, line 67-68: In particular, TZDs, pioglitazone, and WY14643.... Note: pioglitazone is a TZD.  This is not clear from the sentence as written.

2. Page 3, line 89: Corneocytes are normally anucleated, not nucleated.  Nucleated corneocytes would be a pathological sign (parakeratosis). 

3. Page 3: line 110-111. Germline knockout of Pparg is embryonic lethal due to placental deficiency, the studies referred to here involved mice lacking only epidermal Pparg.  

  Comments on the Quality of English Language

There are multiple instances in which words or syntax could be altered to improve readability.

Author Response

Reviewer 1

This review by Briganti et al summarizes multiple studies that indicate an important role for PPARgamma in skin disease and physiology.  After briefly describing the role of PPARgamma as a ligand-activated nuclear receptor, the review discusses the role of PPARgamma in normal skin physiology, including barrier function, sebaceous gland development and corneocyte lipid synthesis.  The review then discusses the role of PPARgamma in skin inflammation, then follows this with a discussion of PPARgamma in skin diseases, particularly inflammatory skin disorders, aging and cancer.  Overall, the review provides a very nice summary of the evidence linking PPARgamma activity in skin biology and common skin diseases.  There are several issues that should be addressed:

  1. Much of the review focuses on the ability of PPARgamma to act as an anti-inflammatory agent.  This anti-inflammatory activity is likely dependent on the ability of PPARgamma to transrepress pro-inflammatory nuclear transcription factors.  While this transrepressive activity is briefly mentioned, it would be useful to include a greater discussion of the mechanisms of transrepression.  A second figure, similar to Fig 1 that highlights transrepressive mechanisms might also be useful. 

We thank the Reviewer for the suggestion. As suggested, we added some information in the Introduction section and, to better clarify the different PPAR regulatory mechanisms, we improved Figure 1 by adding also the transrepressive and protein-protein mechanisms.

  1. Page 5 discusses the effects of PPARgamma on macrophages.  It would be useful to include studies by Hong Du and Cong Yan that show that dominant negative PPARgamma expression in macrophages drives MDSC formation, immunosuppression and tumorigenesis [Blood. 2012, 119(1):115-26].  This study is relevant to both macrophage biology as well as the later discussion of PPARgamma in cutaneous malignancy. 

We added this aspect in the text: Page 7 lines 221-226: “Furthermore, overexpression of dominant-negative PPARgamma in macrophages results in upregulation of pro-inflammatory cytokines/chemokines and expansion of myeloid-derived suppressor cells (MDSCs), leading to inflammation, immunosuppression and tumorigenesis  [53]. These data suggest that the PPARgamma signaling pathway and its downstream gene products are essential for controlling chronic inflammation (especially MDSC homeostasis) and tumorigenesis”.

  1. Page 5, lines 200-2001. "PPARg is a key regulator of the functional maturation of dendritic cells (DC), driving the induction of immunogenic T cell responses rather than immune tolerance".  Yet the next sentence would suggest the opposite: "Pharmacological modulation of PPARg signaling in murine DC ...... significantly inhibited the ability of DC to prime naive CD4+ Tc cells in vitro [91]." The study referenced for this statement also indicates that adoptive transfer of PPARg-activated DCs induced CD4+ T cell anergy.  How does this indicate that PPARgamma drives the ability of DCs to induce immunogenic T cell responses?

We thank the reviewer for his careful review on this point. The misunderstanding was generated by a mistake we made in writing the sentence that contradicted the next one. We wrote rather than instead of versus giving a completely different meaning to the sentence. We have corrected the sentence accordingly: Page 7 lines  227-231 “PPARγ is a key regulator of the functional maturation of dendritic cells (DC), driving the induction of immunogenic T cell responses versus immune tolerance. Klotz et al demonstrated that pharmacological modulation of PPARγ signaling in murine DC reduced the expression of costimulatory molecules and IL-12 associated with the maturation process and significantly inhibited the ability of DC to prime naive CD4+ T cells in vitro [54]”.

The study referenced for this statement also indicates that adoptive transfer of PPARg-activated DCs induced CD4+ T cell anergy.  How does this indicate that PPARgamma drives the ability of DCs to induce immunogenic T cell responses? 

To explain this observation made by the reviewer, we have added the following sentences to the text, again taking into account the fact that the erroneous sentence above had led to misunderstandings about the role of PPARγ on the function of DCs: Page 7 lines  232-236 In addition, CD4+ T cells primed by PPARγ-activated DCs failed to express Th1 and Th2 cytokines and did not respond to further T cell receptor-mediated stimulation with secondary clonal expansion [54]. The authors proposed that PPARγ controls DC function in a complex manner that allows survival of Ag-reactive CD4+ T cells, but induces CD4+ T cell anergy instead of immunity upon secondary Ag encounter.

 Minor issues:

  1. Page 3, line 67-68: In particular, TZDs, pioglitazone, and WY14643.... Note: pioglitazone is a TZD.  This is not clear from the sentence as written.

We apologize to the Reviewer for the unclear sentence. We agree with the comment and amended the sentence as follows:  “In particular, TZDs, such as pioglitazone, and WY14643 are some of the receptor ligands that have demonstrated promise in the management of a range of skin conditions, including psoriasis, atopic dermatitis, and skin cancers.”

  1. Page 3, line 89: Corneocytes are normally anucleated, not nucleated.  Nucleated corneocytes would be a pathological sign (parakeratosis). 

We apologize for the typo error; we corrected it in the revised version.

  1. Page 3: line 110-111. Germline knockout of Pparg is embryonic lethal due to placental deficiency, the studies referred to here involved mice lacking only epidermal Pparg.  

We thank the Reviewer for the comment. We have corrected the sentence accordingly.

Reviewer 2 Report

Comments and Suggestions for Authors

The authors present a systematic review study aiming to describe the current knowledge of PPARγ in skin physiopathology, highlighting it as the main therapeutic target in skin inflammatory disorders and cancer.

I would like to raise the following concerns.

Although Figure 1 illustrates PPARs signaling and functions, a summary Table highlighting PPARγ (and its related effect size) as the main current cutaneous therapeutic target in skin inflammatory disorders and cancer might be beneficial for readers.

Author Response

Reviewer 2

The authors present a systematic review study aiming to describe the current knowledge of PPARγ in skin physiopathology, highlighting it as the main therapeutic target in skin inflammatory disorders and cancer.

I would like to raise the following concerns.

Although Figure 1 illustrates PPARs signaling and functions, a summary Table highlighting PPARγ (and its related effect size) as the main current cutaneous therapeutic target in skin inflammatory disorders and cancer might be beneficial for readers.

We thank the Reviewer for the suggestion. We have added Table 1 in the paragraph “Conclusions” to highlight the PPARγ effects in each mentioned disease, indicating also the most tested molecules. We hope that this will improve the quality of the Review.

Reviewer 3 Report

Comments and Suggestions for Authors

This manuscript systematically describes the skin  physiopathology of PPARγ and is a high-quality review article. This paper is well writing and summary. For review articles, however, word descriptions are too noisy and difficult to arouse readers’ interest. If more pictures are added, it will be easier for readers to quickly understand the information. As such, I recommend this manuscript to be published after appropriate revision as below:

1.     Figure 1A. The content is too simple, and more information is needed, for example, which downstream genes will be affected by co-activators and co-repressors?

2.     Can the structural differences or amino acid sequence differences in the ligand binding pocket between different PPAR subtypes (α, β/δ, and γ) be shown?

3.     Please list the structures of representative PPARy molecules related to skin pathophysiology;

4.     There should be a diagram for PPARγ and each skin disease, with detailed annotations of the corresponding mechanisms, which will make it easier for readers to understand;

5.     Part “5.5 PPARγ and skin aging” and “5.6 PPARγ and skin cancer” should become “5. PPARγ and skin aging” and “6. PPARγ and skin cancer”.

Author Response

Reviewer 3

This manuscript systematically describes the skin  physiopathology of PPARγ and is a high-quality review article. This paper is well writing and summary. For review articles, however, word descriptions are too noisy and difficult to arouse readers’ interest. If more pictures are added, it will be easier for readers to quickly understand the information. As such, I recommend this manuscript to be published after appropriate revision as below:

  1. Figure 1A. The content is too simple, and more information is needed, for example, which downstream genes will be affected by co-activators and co-repressors?

We thank the Reviewer for the suggestion. Given the large number of genes directly regulated by the three PPAR isoforms, we thought it would be better to include the different regulatory mechanisms controlled by PPARs (transrepressive and protein-protein mechanisms), rather than give just a few examples of downstream genes. This, together with the table in Figure 1B, may provide an overall view of the regulatory mechanisms and effects of the activation of these nuclear receptors.

  1. Can the structural differences or amino acid sequence differences in the ligand binding pocket between different PPAR subtypes (α, β/δ, and γ) be shown?

We have added Figure 2 to show structural differences between the three PPAR isoforms. We have obtained the different PPAR 3D structures from the Protein Data Bank (PDB) with PDB entries 1I7G (PPARα), 3TKM (PPARβ/δ) and 2F4B (PPARγ). We have also included, in the Figure 2B, the residues alignment of hPPARα (Q07869), hPPARβ/δ (Q03181) and hPPARγ (P37231) using Uni-Prot Align. Accordingly, we have added a sentence in the “Introduction” and the “Methods” section to illustrate the tools used.

  1. Please list the structures of representative PPARy molecules related to skin pathophysiology;
  2. There should be a diagram for PPARγ and each skin disease, with detailed annotations of the corresponding mechanisms, which will make it easier for readers to understand;

We thank the Reviewer for the suggestion. We have added Table 1 in the paragraph “Conclusions” to highlight the PPARγ effects in each mentioned disease, indicating also the most tested molecules. We also added Figure 3 in the same paragraph showing the structures of representative PPARγ activators. Accordingly, we added the “Methods” section to illustrate the tools used to draw the chemical structures. We hope that this will improve the quality of the Review.

  1. Part “5.5 PPARγ and skin aging” and “5.6 PPARγ and skin cancer” should become “5. PPARγ and skin aging” and “6. PPARγ and skin cancer”.

We agree with the Reviewer and changed the order number of paragraphs.

Round 2

Reviewer 2 Report

Comments and Suggestions for Authors

All the concerns have been answered.

Reviewer 3 Report

Comments and Suggestions for Authors

Accept in present form